# A Sampling-Free Approximation of Gaussian Variational Auto-Encoders

## Abstract

We propose a sampling-free approximate formulation of Gaussian variational auto-encoders. Instead of computing the loss via stochastic sampling, we propagate the Gaussian distributions from the latent space into the output space. As computing the exact probability is intractable, we propose to locally approximate the decoder network by its Taylor series. We demonstrate that this approximation allows us to approximate the Gaussian variational auto-encoder training objective in closed form. We evaluate the proposed method on the MNIST, the Omniglot, the CelebA, and the 3D Chairs data sets. We find that our sampling-free approximation performs better than its sampling counterpart on the Frechet inception distance and on par on the estimated marginal likelihood.

## 1 Introduction

Variational Auto-Encoders (VAE) allow generating data, mapping data into a latent space, and modifying data by perturbing it in a controlled manner in the latent space [1, 2]. This enables them to generate and modify images [3, 4], speech [5], and molecular structures [6], or detect outliers in distributions [7]. VAEs extend the concept of regular auto-encoders by modeling the latent space as a distribution and with a distributional constraint that allows for a distributional latent space. Thus, they enforce all samples drawn from the prior to be representative for the data set. For this, VAEs predict a distribution through the encoder, sample from this distribution, and propagate the sample through the decoder. During training, they constrain the predicted distribution with respect to the prior, and require that the reconstructions produced by the decoder are correct. To allow for random sampling in the context of gradient descent, Gaussian VAEs employ the sampling reparameterization trick [8], which enables backpropagation through the sampling operation. In fact, the sampling reparameterization trick and VAEs have received a lot of attention in research [3, 9–11].

Recently, research has also started to look into techniques to compute the posterior and marginal distribution in an exact analytical solution without the need for sampling. In a sampling-free formulation, during training, the loss is not computed via stochastic sampling but derived by propagating the distributions from the latent space to the output space. However, computing the exact likelihood is in many cases intractable. Two approaches have recently been proposed to address this challenging problem. Balestriero *et al.* [12] propose to compute the likelihood for small neural networks by integration over piecewise continuous polytopes. Lucas *et al.* [13] formulate an analytical sampling-free *linear* VAE, which can be efficiently computed in closed form.

In this context, the following work proposes a sampling-free approximate formulation for VAEs that allows sampling-free training of VAEs. To this end, we approach the problem of intractability by approximating the probability distributions instead of computing them exactly. This approximation is done by local Taylor series expansion of locally affine (ReLU-activated) neural networks. This corresponds to the local linearization of the respective neural network and allows for a *local* linearization of the decoder of the Gaussian VAE. We propose to use the local linearization to estimate the (co-)variances for each data point, while the median of the output distribution is computed exactly. Locally approximating neural networks by their Taylor series around each individual data point enables us to express the covariance matrix of the probability distribution in terms of the Jacobian of the neural network. Furthermore, as computing the covariance matrix explicitly is very expensive, the proposed formulation allows computing the likelihood without explicitly computing the covariance matrix nor its inverse. This allows us to compute the training objective without sampling at a reason-

able additional cost for standard architectures, specifically, at the cost of evaluating the Jacobian of the decoder. Note that the local Taylor series expansion or local linearization of the neural network decoder function does neither correspond to a global linearization of the neural network decoder nor a global approximation via its Maclaurin series.

We evaluate the proposed method on the CelebA [14], the 3D Chairs [15], and the MNIST [16] data sets. When considering the number of samples necessary to estimate a covariance matrix of equivalent quality wrt. the Frobenius norm via stochastic sampling, our approximation is (on average) equivalent to more than 235 samples for a deep network on the CelebA data set. Overall, the proposed approach achieves competitive results compared to sampling Gaussian VAEs. Especially on the Frechet Inception Distance (FID) [17], the sampling-free VAE performs better than the sampling VAE, while the estimated marginal likelihoods between both methods are on par. This demonstrates that local linearization is a good approximation and does not damage performance in comparison to a regular sampling VAE. Following the setup by Lucas *et al.* [13], the sampling-free VAE leads to smaller $(\epsilon, \delta)-$posterior collapse than its sampling counterpart.

We summarize the contributions of this work as follows:

- We present a novel sampling-free approximation for training Gaussian VAEs. In contrast to exact computation, it requires only a small overhead compared to its sampling counterpart, and reaches a less noisy approximation of the training objective than sampling.
- We show that the sampling-free Gaussian VAE approximation achieves competitive performance on the MNIST, Omniglot, CelebA, and 3D Chairs data sets compared to sampling VAE methods. Also, the sampling-free formulation also achieves a better FID on CelebA when provided with the same computational budget as its sampling counterpart.
- We find that sampling-free VAEs are more robust against posterior collapse than sampling VAEs.

The full implementation as well as experiments will be publicly available upon publication.

## 2 RELATED WORK

Balestriero *et al.* [12] present an approach to compute the exact likelihood for small yet deep neural networks using the continuous piecewise affinity of respective neural networks to compute the exact likelihood by partitioning the network and integrating over the polytopes. Based on this, an analytical Expectation-Maximization algorithm is derived that enables gradient-free learning. The method is empirically validated by training a three layer deep generative network with 8 and 16 hidden units and with one latent dimension on the number 4 from the MNIST data set, where they demonstrate that an exact sampling-free VAE is superior to the stochastic sampling VAE. However, as the exact analytical formulation of deep neural networks has a computational complexity that is exponential in the size of the neural network, it is intractable for larger networks. Lucas *et al.* [13] examine sampling-free linear VAEs, which by the nature of their linearity allow for efficient exact computation of the likelihood and are able to show that the linear VAE recovers probabilistic principal component analysis (pPCA) [18]. They employ the sampling-free linear VAE for understanding posterior collapse in VAEs and apply it to the MNIST data set.

Ghosh *et al.* [19] propose regularized deterministic auto-encoders, to learn a smooth latent space without sampling via different regularization schemes. They regularize their auto-encoders through weight decay, the gradient norm of the decoder and/or spectral normalization, which all bound the Lipschitz continuity of the decoder. Our work (implicitly) also includes a kind of gradient regularization of the decoder. However, in our work, this arises from approximating the marginal log-likelihood via local Taylor approximations of the decoder.

Tolstikhin *et al.* [20] propose Wasserstein Auto-Encoders (WAEs), which minimize the Wasserstein distance between the model distribution and the target distribution. Kolouri *et al.* [21] propose Sliced Wasserstein Auto-Encoders (SWAEs), which regularize the auto-encoders using the sliced / marginal Wasserstein distance between the distribution of the encoded training samples and a predefined sampleable distribution. Note that both WAEs and SWAEs require sampling [20–22]. Knop *et al.* [22] propose Cramer-Wold Auto-Encoders (CWAEs), a sampling-free distributionally regularized Auto-Encoder. They introduce the Cramer-Wold distance, which has a simple analytical formula for computing normality of high-dimensional samples. By regularizing their auto-encoders via the

Cramer-Wold distance, they can evaluate their training objective in closed form. Note that CWAEs are not VAEs as, for VAEs, the distance to the true distribution is measured using KL divergence under the latent variable model [22]. Instead, the CWAE is a different kind of generative model that was designed to have a tractable sampling-free objective.

Burda *et al.* [23] introduce importance weighted sampling for VAEs by using multiple samples to have a tighter variational lower bound. Huang *et al.* [24] propose hierarchical importance weighted auto-encoders reducing the number of redundant samples and introducing a hierarchical structure to induce correlation. Dieng *et al.* [25] use importance weighted VAEs using an EM-based algorithm employing moment matching. Roeder *et al.* [26] propose a low-variance gradient estimator for VAEs. Park *et al.* [27] enrich the posterior by applying the Laplace approximation to VAEs, which enables modeling a more expressive full-covariance Gaussian posterior. Tomczak *et al.* [28] also produce an improved posterior distribution through a Householder transformation-based volume-preserving flow. Nielsen *et al.* [29] bridge the gap between normalizing flows [30] and VAEs by preseting SurVAE Flows, which use surjective transformation which are deterministic in one direction and stochastic / sampling in the reverse direction. Morrow *et al.* [31] present VAEs with normalizing flow decoders.

Taylor series approximations [32] are a ubiquitous tool in variational inference (VI), e.g., in the delta or the Laplace method [33, 34]. VI methods, such as the delta method or the Laplace method require solving an optimization problem to obtain an approximation of an output distribution. In contrast, our method uses the Taylor approximation to produce a closed-form estimate of the training objective and the only optimization is the training of the VAE.

A phenomenon frequently appearing in VAEs is posterior collapse, i.e., the posterior distribution (produced by an encoder) is close to the prior distribution. This leads to a reduced expressivity of the VAE's latent space. To tackle this, Kingma *et al.* [8] proposed to constrain the KL divergence such that it is only active if the KL divergence is above a threshold. Other works used KL annealing [35, 36] or constrained the posterior to have a minimum distance to the prior [37]. Lucas *et al.* [13] defined the $(\epsilon, \delta)-$posterior collapse in order to reliably quantify posterior collapse in VAEs.

## 3 METHOD

We begin this section by introducing the conventional data generating process of VAEs. After that, we briefly motivate and derive the evidence lower bound (ELBO) and present our sampling-free local approximation of the ELBO including an efficient way of computing it. Following this, we discuss a method for stabilizing training and investigate the balance between reconstruction quality and variance of reconstructions.

### 3.1 GAUSSIAN VARIATIONAL AUTO-ENCODERS

We begin by stating the assumptions about the data generating process. Note that this process does not differ from the conventional data generating process of VAEs [1]. For easier reference, we follow the notation of Kingma and Welling [9]. The goal of VAEs is to approximate the true distribution of the data $p^*(\mathbf{x})$ with a distribution parameterized via a model $p_\theta(\mathbf{x}) \approx p^*(\mathbf{x})$. Here,

$$p(\mathbf{z}) = \mathcal{N}(\mathbf{z}; 0, \boldsymbol{I}) \tag{1}$$

$$p_\theta(\mathbf{x}|\mathbf{z}) = \mathcal{N}(\mathbf{x}; f_\theta(\mathbf{z}), c\boldsymbol{I}) \quad \text{for some } c > 0 \tag{2}$$

$$p_\theta(\mathbf{x}) = \int p_\theta(\mathbf{z}) p_\theta(\mathbf{x}|\mathbf{z}) \, d\mathbf{z} \,. \tag{3}$$

$\mathbf{z}$ denotes an $m$-dimensional vector referring to the hidden / latent space, $\mathbf{x}$ denotes an $n$-dimensional vector referring to the input / output space, $f_\theta$ is the decoder / generator network, and $c$ captures the observation uncertainty. $\mathcal{N}$ denotes a multivariate Gaussian distribution and $\boldsymbol{I}_m$ the $m \times m$ identity matrix. To approximate the intractable $p_\theta(\mathbf{z}|\mathbf{x})$, VAEs include an encoder network $g_\phi$ and a network that computes the encoder's uncertainty $h_\phi$ (in the form of a diagonal covariance matrix) such that:

$$q_\phi(\mathbf{z}|\mathbf{x}) \approx p_\theta(\mathbf{z}|\mathbf{x}) \qquad \text{where} \quad q_\phi(\mathbf{z}|\mathbf{x}) = \mathcal{N}(\mathbf{z}; g_\phi(\mathbf{x}), h_\phi(\mathbf{x})). \tag{4}$$

The training objective is to maximize the log-likelihood of our data $\log p_\theta(\mathbf{x})$ by optimizing $\theta$ and $\phi$:

$$\log p_\theta(\mathbf{x}) = \mathbb{E}_{q_\phi(\mathbf{z}|\mathbf{x})} \left( \log \frac{p_\theta(\mathbf{x}|\mathbf{z})p(\mathbf{z})}{q_\phi(\mathbf{z}|\mathbf{x})} \right) + \mathbb{E}_{q_\phi(\mathbf{z}|\mathbf{x})} \left( \log \frac{q_\phi(\mathbf{z}|\mathbf{x})}{p_\theta(\mathbf{z}|\mathbf{x})} \right) \qquad \text{(cf. [9])} \tag{5}$$

$$= \mathbb{E}_{q_\phi(\mathbf{z}|\mathbf{x})} \left( \log p_\theta(\mathbf{x}|\mathbf{z}) \right) - \mathbb{E}_{q_\phi(\mathbf{z}|\mathbf{x})} \left( \log \frac{q_\phi(\mathbf{z}|\mathbf{x})}{p(\mathbf{z})} \right) + \mathbb{E}_{q_\phi(\mathbf{z}|\mathbf{x})} \left( \log \frac{q_\phi(\mathbf{z}|\mathbf{x})}{p_\theta(\mathbf{z}|\mathbf{x})} \right) \tag{6}$$

$$= \underbrace{\mathbb{E}_{q_\phi(\mathbf{z}|\mathbf{x})} \left( \log p_\theta(\mathbf{x}|\mathbf{z}) \right) - D_{\mathrm{KL}}(q_\phi(\mathbf{z}|\mathbf{x}), p(\mathbf{z}))}_{\text{ELBO}} + \underbrace{D_{\mathrm{KL}}(q_\phi(\mathbf{z}|\mathbf{x}), p_\theta(\mathbf{z}|\mathbf{x}))}_{\text{not computable but non-negative}}. \tag{7}$$

The last term is not computable, because $p_\theta(\mathbf{z}|\mathbf{x})$ is intractable, and hence the log-likelihood $\log p_\theta(\mathbf{x})$ cannot be computed exactly. However, since a KL-divergence is non-negative, the first two terms yield a lower bound for $\log p_\theta(\mathbf{x})$, the so-called evidence lower bound (ELBO) [1, 9, 38]. The ELBO is the substitute objective that allows training VAEs.

$$\mathrm{ELBO} = \log p_\theta(\mathbf{x}) - D_{\mathrm{KL}}(q_\phi(\mathbf{z}|\mathbf{x}), p_\theta(\mathbf{z}|\mathbf{x})) \ \leq \ \log p_\theta(\mathbf{x}) \tag{8}$$

The KL-Divergence $D_{\mathrm{KL}}(q_\phi(\mathbf{z}|\mathbf{x}), p(\mathbf{z}))$ can be computed in closed form. However, the computation of $\mathbb{E}_{q_\phi(\mathbf{z}|\mathbf{x})}(\log p_\theta(\mathbf{x}|\mathbf{z}))$ is generally intractable and is therefore usually approximated via stochastic sampling from the latent distribution $q_\phi(\mathbf{z}|\mathbf{x})$. In the stochastic sampling approach, a data point $\mathbf{x}$ is propagated through $g_\phi$ and $h_\phi$, an element $\mathbf{z}$ is sampled from $\mathcal{N}(\mathbf{z}; g_\phi(\mathbf{x}), h_\phi(\mathbf{x}))$, and propagated through $f_\theta$, which approximates the output distribution (including the observation uncertainty) $\mathcal{N}(\mathbf{x}; f_\theta(\mathbf{z}), c\mathbf{I})$ where $\mathbf{z} \sim \mathcal{N}(\mathbf{z}; g_\phi(\mathbf{x}), h_\phi(\mathbf{x}))$. Since the observation uncertainty is an isotropic normal distribution, the sample log-likelihood $\log p_\theta(\mathbf{x}|\mathbf{z})$ reduces to

$$\log p_\theta(\mathbf{x}|\mathbf{z}) = -\frac{1}{2c} \left( \|f_\theta(\mathbf{z}) - \mathbf{x}\|^2 \right) - \frac{n}{2} \log(2\pi c) \tag{9}$$

which is, apart from the normalization constant, a mean squared error [1].

## 3.2 SAMPLING-FREE APPROXIMATION

At this point, we introduce the proposed sampling-free approach. In the sampling-free approach, a data point $\mathbf{x}$ is propagated through $g_\phi$ and $h_\phi$, and then the obtained distribution $\mathcal{N}(\mathbf{z}; g_\phi(\mathbf{x}), h_\phi(\mathbf{x}))$ is (without sampling) approximately propagated through $f_\theta$, which yields the output distribution $\mathcal{N}(\mathbf{x}; f_\theta(\mathbf{z}), c\mathbf{I})$ where $\mathbf{z} \sim \mathcal{N}(\mathbf{z}; g_\phi(\mathbf{x}), h_\phi(\mathbf{x}))$. The problem is that computing the exact transformation of the distribution $\mathcal{N}(\mathbf{z}; g_\phi(\mathbf{x}), h_\phi(\mathbf{x}))$ by $f_\theta$ is intractable (for non-trivial $f_\theta$) because it can have an exponential complexity. The largest analytical sampling-free approach so far was able to operate on 24 hidden neurons [12]. This is because, for the exact computation, every linear region of $f_\theta$ has to be computed separately. To simplify notation, in the following, we drop subscripts $\theta$ and $\phi$ for $f, g, h$.

To make this problem tractable, we approximate the neural network decoder via a local Taylor series expansion, which simplifies computing the output distribution. For that, we define the Taylor series of the decoder $f$ around $g(\mathbf{x})$ for each data point $\mathbf{x}$ as $\hat{f}_{g(\mathbf{x})}$. We display the Taylor series of an example neural network function in Figure A.1 in the supplementary material. Notably, the function and its Taylor series are equal near this point $g(\mathbf{x})$, which is also the point of largest density of the distribution $\mathcal{N}(\mathbf{z}; g(\mathbf{x}), h(\mathbf{x}))$. Due to the continuous piece-wise affinity of $f$, we only need to compute the first two terms of the Taylor expansion, as the remaining terms are zero. Thus,

$$\hat{f}_{g(\mathbf{x})}(g(\mathbf{x}) + \epsilon) = f(g(\mathbf{x})) + \frac{d\,f(g(\mathbf{x}))}{d\,g(\mathbf{x})} \cdot \epsilon = f(g(\mathbf{x})) + \mathbf{J}_{f(g(\mathbf{x}))} \cdot \epsilon \tag{10}$$

where $\mathbf{J}_{f(g(\mathbf{x}))}$ is the Jacobian of $f$ at point $g(\mathbf{x})$, in the following abbreviated as $\mathbf{J}$. This approximation is exact if $\epsilon$ is small enough, such that it stays inside the linear region of $f$ in which $g(\mathbf{x})$ is located. In Section 4.1, we empirically demonstrate that the Taylor series delivers a good approximation for the neural network decoders in VAEs. For neural networks that are not piece-wise affine, like the tanh networks used in the experiments in Section 4.3, we also use Equation 10, which in this case is the first-order Taylor expansion. Using the Taylor series for each point $\hat{f}_{g(\mathbf{x})}$ we can approximate the output distribution as follows:

$$\hat{f}_{g(\mathbf{x})}(\mathbf{z}) \approx f(\mathbf{z}) \qquad \text{where } \mathbf{z} \sim \mathcal{N}(\mathbf{z}; g(\mathbf{x}), h(\mathbf{x})) \tag{11}$$

As the family of Gaussian distributions is closed under affine transformations and summation [39]:

$$\hat{f}_{g(\mathbf{x})}(\mathbf{z}) \sim \mathcal{N}\left(\mathbf{x}'; f(g(\mathbf{x})), \mathbf{J}\, h(\mathbf{x})\, \mathbf{J}^\top\right) \qquad \text{where } \mathbf{z} \sim \mathcal{N}(\mathbf{z}; g(\mathbf{x}), h(\mathbf{x})) \qquad (12)$$

and including the observation uncertainty

$$\hat{f}_{g(\mathbf{x})}(\mathbf{z}) + \mathbf{o} \sim \mathcal{N}\left(\mathbf{x}'; f(g(\mathbf{x})), \underbrace{\mathbf{J}\, h(\mathbf{x})\, \mathbf{J}^\top + c\mathbf{I}}_{=:\, \boldsymbol{\Sigma}_{\mathbf{x}}}\right) \qquad \text{where } \mathbf{o} \sim \mathcal{N}(\mathbf{o}; 0, c\mathbf{I}). \qquad (13)$$

This leads to a sampling-free approximation of the log-likelihood

$$\mathbb{E}_{q_\phi(\mathbf{z}|\mathbf{x})} \log p_\theta(\mathbf{x}|\mathbf{z}) \approx \log\left((2\pi)^{-\frac{n}{2}} \det(\boldsymbol{\Sigma}_{\mathbf{x}})^{-\frac{1}{2}} \exp\left(-\tfrac{1}{2}(f(g(\mathbf{x})) - \mathbf{x})^\top \boldsymbol{\Sigma}_{\mathbf{x}}^{-1}(f(g(\mathbf{x})) - \mathbf{x})\right)\right)$$

$$= -\tfrac{n}{2}\log(2\pi) - \tfrac{1}{2}\log\det(\boldsymbol{\Sigma}_{\mathbf{x}}) - \tfrac{1}{2}(f(g(\mathbf{x})) - \mathbf{x})^\top \boldsymbol{\Sigma}_{\mathbf{x}}^{-1}(f(g(\mathbf{x})) - \mathbf{x}) \qquad (14)$$

At this point, it is necessary to resolve $\boldsymbol{\Sigma}_{\mathbf{x}}^{-1}$ and $\det(\boldsymbol{\Sigma}_{\mathbf{x}})$.

**Lemma 1** (Woodbury matrix identity). *[40]. The inverse of a rank-$m$ correction of a matrix can be computed in terms of the inverse of the original matrix as*

$$(\boldsymbol{A} + \boldsymbol{U}\boldsymbol{V})^{-1} = \boldsymbol{A}^{-1} - \boldsymbol{A}^{-1}\boldsymbol{U}(\boldsymbol{I}_m + \boldsymbol{V}\boldsymbol{A}^{-1}\boldsymbol{U})^{-1}\boldsymbol{V}\boldsymbol{A}^{-1}. \qquad (15)$$

Since $\boldsymbol{\Sigma}_{\mathbf{x}} = c\boldsymbol{I} + \mathbf{J}\, h(\mathbf{x})\, \mathbf{J}^\top$, and $h(\mathbf{x})$ is a diagonal matrix, we can set $\boldsymbol{U} = \mathbf{J}\sqrt{h(\mathbf{x})}$ such that $\boldsymbol{U}\boldsymbol{U}^\top = \boldsymbol{U}\boldsymbol{V} = \mathbf{J}\, h(\mathbf{x})\, \mathbf{J}^\top$. As the computation of the covariance via the Jacobian directly produces a low-rank decomposition of the covariance matrix and because $m \ll n$, this allows for a fast and space-efficient computation. Thus,

$$\begin{aligned}
\boldsymbol{\Sigma}_{\mathbf{x}}^{-1} \quad &= \left(c\boldsymbol{I} + \mathbf{J}h(\mathbf{x})\mathbf{J}^\top\right)^{-1} & &= \left(c\boldsymbol{I} + \boldsymbol{U}\boldsymbol{U}^\top\right)^{-1} \\
&= \tfrac{1}{c}\boldsymbol{I} - \tfrac{1}{c}\boldsymbol{I}\boldsymbol{U}(\boldsymbol{I}_m + \boldsymbol{U}^\top\tfrac{1}{c}\boldsymbol{I}\boldsymbol{U})^{-1}\boldsymbol{U}^\top\tfrac{1}{c}\boldsymbol{I} & &= \tfrac{1}{c}\boldsymbol{I} - \tfrac{1}{c^2}\boldsymbol{U}(\boldsymbol{I}_m + \tfrac{1}{c}\boldsymbol{U}^\top\boldsymbol{U})^{-1}\boldsymbol{U}^\top & (16) \\
&= \tfrac{1}{c}\left(\boldsymbol{I} - \tfrac{1}{c}\boldsymbol{U}(\boldsymbol{I}_m + \tfrac{1}{c}\boldsymbol{U}^\top\boldsymbol{U})^{-1}\boldsymbol{U}^\top\right) & &= \tfrac{1}{c}\left(\boldsymbol{I} - \boldsymbol{U}(c\boldsymbol{I}_m + \boldsymbol{U}^\top\boldsymbol{U})^{-1}\boldsymbol{U}^\top\right)
\end{aligned}$$

which only requires inverting an $m \times m$ matrix (innermost parentheses).

**Lemma 2** (Matrix determinant lemma). *[41]. The determinant of a rank-$m$ correction of a matrix can be computed in terms of the inverse and determinant of the original matrix as*

$$\det(\boldsymbol{A} + \boldsymbol{U}\boldsymbol{V}) = \det(\boldsymbol{I}_m + \boldsymbol{V}\boldsymbol{A}^{-1}\boldsymbol{U})\det(\boldsymbol{A}) \qquad (17)$$

With $\boldsymbol{U}$ defined as above, we can see that

$$\begin{aligned}
\det(\boldsymbol{\Sigma}_{\mathbf{x}}) \quad &= \det\left(c\boldsymbol{I} + \mathbf{J}h(\mathbf{x})\mathbf{J}^\top\right) & &= \det\left(c\boldsymbol{I} + \boldsymbol{U}\boldsymbol{U}^\top\right) \\
&= \det\left(\boldsymbol{I}_m + \boldsymbol{U}^\top\tfrac{1}{c}\boldsymbol{I}\boldsymbol{U}\right)\det(c\boldsymbol{I}) & &= \det\left(\boldsymbol{I}_m + \tfrac{1}{c}\boldsymbol{U}^\top\boldsymbol{U}\right)c^n
\end{aligned} \qquad (18)$$

As we can now compute $\boldsymbol{\Sigma}_{\mathbf{x}}^{-1}$ and $\det(\boldsymbol{\Sigma}_{\mathbf{x}})$ very efficiently, we can extend the log-likelihood as

$$\begin{aligned}
\mathbb{E}_{q_\phi(\mathbf{z}|\mathbf{x})} \log p_\theta(\mathbf{x}|\mathbf{z}) &\approx -\tfrac{n}{2}\log(2\pi) - \tfrac{1}{2}\log\det(\boldsymbol{\Sigma}_{\mathbf{x}}) - \tfrac{1}{2}(f(g(\mathbf{x})) - \mathbf{x})^\top \boldsymbol{\Sigma}_{\mathbf{x}}^{-1}(f(g(\mathbf{x})) - \mathbf{x}) \\
&= -\tfrac{n}{2}\log(2\pi) - \tfrac{1}{2}\log\det(\boldsymbol{I}_m + \tfrac{1}{c}\boldsymbol{U}^\top\boldsymbol{U}) - \tfrac{n}{2}\log(c) \\
&\quad \underbrace{-\tfrac{1}{2}(f(g(\mathbf{x})) - \mathbf{x})^\top\tfrac{1}{c}\left(\boldsymbol{I} - \boldsymbol{U}(c\boldsymbol{I}_m + \boldsymbol{U}^\top\boldsymbol{U})^{-1}\boldsymbol{U}^\top\right) \cdot (f(g(\mathbf{x})) - \mathbf{x})} \qquad (19)
\end{aligned}$$
$$= -\tfrac{1}{2c}(f(g(\mathbf{x}))-\mathbf{x})^\top(f(g(\mathbf{x}))-\mathbf{x}) + \tfrac{1}{2c}((f(g(\mathbf{x}))-\mathbf{x})^\top\boldsymbol{U})(c\boldsymbol{I}_m + \boldsymbol{U}^\top\boldsymbol{U})^{-1}\cdot(\boldsymbol{U}^\top(f(g(\mathbf{x}))-\mathbf{x}))$$

Here, as the inverse of the covariance matrix ($\boldsymbol{\Sigma}_{\mathbf{x}}^{-1}$ of size $n \times n$) does not need to be computed explicitly and the factorization of the covariance can be multiplied with the error, only operations on small matrices need to be performed. Specifically, it requires one multiplication of a vector of size $n$ with a matrix of size $m \times n$, the inversion of an $m \times m$ matrix, and the multiplication of vectors from both sides with an $m \times m$ matrix. This significantly reduces the computation cost.

Based on these insights, we can write the ELBO as

$$\begin{aligned}
\text{ELBO} &= \mathbb{E}_{q_\phi(\mathbf{z}|\mathbf{x})}\left(\log p_\theta(\mathbf{x}|\mathbf{z})\right) - D_{\text{KL}}(q_\phi(\mathbf{z}|\mathbf{x}), p(\mathbf{z})) \qquad (20) \\
&\approx -\tfrac{n}{2}\log(2\pi c) - \tfrac{1}{2}\log\det(\boldsymbol{I}_m + \tfrac{1}{c}\boldsymbol{U}^\top\boldsymbol{U}) - \tfrac{1}{2c}(f(g(\mathbf{x})) - \mathbf{x})^\top(f(g(\mathbf{x})) - \mathbf{x}) \\
&\quad + \tfrac{1}{2c}\left((f(g(\mathbf{x})) - \mathbf{x})^\top\boldsymbol{U}\right)(c\boldsymbol{I}_m + \boldsymbol{U}^\top\boldsymbol{U})^{-1}\left(\boldsymbol{U}^\top(f(g(\mathbf{x})) - \mathbf{x})\right) - D_{\text{KL}}(q_\phi(\mathbf{z}|\mathbf{x}), p(\mathbf{z}))
\end{aligned}$$

Since the training objective is to maximize the ELBO, the corresponding loss is $\mathcal{L} = -\,\text{ELBO}$.

### 3.3 STABILIZING TRAINING AND REDUCING BIAS BY REGULARIZATION

In practice, for covariance matrices with large variances, the ratio between the largest and smallest eigenvalue can become extremely large even with Tikhonov regularization (e.g. $10^8$). Thus, training is not feasible because the gradients in some or most directions are vanishingly small. To account for this, we propose a regularization that works independently of the ratio between eigenvalues. For this, we use a mixture between the approximated log-likelihood as described above and the log-likelihood of a conventional auto-encoder (i.e., only considering observation-noise). Specifically, we replace $\mathbf{\Sigma_x^{-1}}$ by $\gamma \cdot \mathbf{\Sigma_x^{-1}} + (1 - \gamma) \cdot (c\mathbf{I})^{-1}$. This regularization allows for an efficient and stable training, even if training with $\mathbf{\Sigma_x^{-1}}$ alone would be vanishingly slow or diverges. This means, we calculate

$$\mathbf{\Sigma}_{\mathbf{x},\gamma}^{-1} \approx \tfrac{1}{c}\mathbf{I} - \gamma\tfrac{1}{c}\mathbf{U}(c\mathbf{I} + \mathbf{U}^\top\mathbf{U})^{-1}\mathbf{U}^\top \quad \text{instead of} \quad \mathbf{\Sigma_x^{-1}} = \tfrac{1}{c}\mathbf{I} - \tfrac{1}{c}\mathbf{U}(c\mathbf{I} + \mathbf{U}^\top\mathbf{U})^{-1}\mathbf{U}^\top \quad (21)$$

where $\gamma$ quantifies the degree to which the variation produced by the decoder is considered. In our experiments, we found that $\gamma \approx 0.99$ avoids vanishingly slow training and divergence due to numerical instabilities. Empirically, we found that reducing $\gamma$ can reduce the bias of our sampling-free approximation. In Figure 1, we train a model using the proposed sampling-free objective with $\gamma$ and evaluate it using the unbiased sampling ELBO. We find that with $\gamma{=}0.95$, the model converges to an ELBO of $-8000$. By reducing $\gamma$, i.e., increasing the regularization, we can increase the ELBO that the model converges to, which means that we improve the model. Setting $\gamma{=}0.67$, the ELBO already converges to $-7500$, and after setting $\gamma{=}0.5$ it converges to $-7400$. We find that $\gamma = 0.5$ works well overall and, thus, we use it as a canonical choice in our experiments.

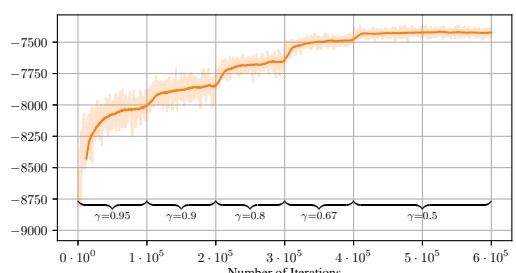

Figure 1: By scheduling $\gamma$, we can observe that increasing the regularization (i.e., reducing $\gamma$) reduces the approximation bias and increases the point of convergence, which improves the model. The model is trained with the biased sampling-free VAE objective on the CelebA data set and evaluated using the unbiased sampling ELBO estimate.

## 4 EXPERIMENTS

### 4.1 QUALITY OF THE TAYLOR SERIES APPROXIMATION

To evaluate the quality of the Taylor series approximation in VAEs, we consider the precision to which variances and covariances are estimated and compare this to different numbers of samples. For this, we use sampling VAEs trained on the CelebA and 3D Chairs data sets as described in more detail in the next section and with a latent dimensionality of 16. We consider two metrics: The Pearson correlation coefficient of the (pixel-wise) output

Table 1: Quality of the Taylor series approximation. For both data sets, we trained a (sampling) VAE and evaluate the Pearson correlation between the true and estimated variances. For the CelebA data set, we also evaluate the Frobenius norm between the true and estimated covariance matrices. We evaluate our approximation and compare it to different numbers of samples.

| Method | | Taylor | Sampling | | | | | | | |
|--------|--------|--------|------|------|------|------|------|------|------|
| #Points | | 1 | 5 | 10 | 25 | 50 | 70 | 100 | 235 |
| CelebA | Var. Corr. | **.967** | .680 | .816 | .912 | .955 | **.966** | .976 | .988 |
| | Cov. Frob. | **297** | 17 446 | 7 779 | 2 951 | 1 456 | 1 033 | 737 | **302** |
| 3D Chairs | Var. Corr. | **.975** | .686 | .819 | .914 | .953 | .966 | **.976** | .988 |

variances and the Frobenius norms of output covariance matrices. Here, we compute the true output (co-)variances via a large number of samples ($15\,000$) and compare it to the (co-)variances produced by our approximation or the (co-)variances produced by a certain number of samples. Results are shown in Table 1. We find that, even in large deep neural networks, we achieve Pearson correlations of $96.7 - 97.5\%$ for estimating the variances, which would require $70 - 100$ samples in the latent space when relying on sampling. On the Frobenius norm of covariance matrices, our approximation is (on average) equivalent to more than 235 stochastic samples for the CelebA data set. Because computing the true covariance matrix is intractable, we estimate it to a precision of $1\%$ of the reported Frobenius norms. For the 3D Chairs data set, we omit estimating the true covariance matrices due to

computational feasibility. In addition, we analyzed the ratio between the variances produced by our sampling-free propagation and the sampling approach and found that across data sets this ratio is around 1.2. This means that the variances are over-estimated by a factor of 1.2 by our approximation. We attribute this to the fact that non-linearities such as ReLU can become constant which is not anticipated by the Taylor approximation.

## 4.2 MNIST, CELEBA, AND 3D CHAIRS

We evaluate the sampling-free VAE on the CelebA data set [14], on the 3D Chairs data set [15], as well as on the MNIST data set [16]. For the CelebA and 3D Chairs experiments, we use the same network architecture as Higgins *et al.* [3] and train it with the Adam optimizer [42] at a learning rate of $10^{-4}$ for $10^6$ iterations. We use $12 - 64$ latent dimensions for CelebA and $8 - 32$ latent dimensions for 3D Chairs. As in previous works on VAEs, we set $c = 0.5$ which corresponds to a standard deviation of the observation uncertainty of $\sigma = \sqrt{0.5}$ for all methods. We use a model for the encoder and decoder with 5 convolutional and 1 fully connected layers each. This is also the architecture that was used in many recent works on sampling VAEs [3, 11] as well as similar to most public implementations. Further details can be found in the supplementary material.

**Qualitative Evaluation** In Figure 2, we show the manifold of a sampling-free VAE in comparison to a sampling VAE. Figure 3 displays qualitative results for CelebA and 3D Chairs. Here, we display reconstruction results as well as traversals in the latent space for 3 images and 2 latent dimensions. In the supplementary material, we qualitatively compare the sampling-free VAE to the sampling VAE, the (sampling) $\beta-$VAE, as well as to a conventional auto-encoder.

Figure 3 shows that the sampling-free VAE can successfully reconstruct and that latent traversals produce meaningful images. For the CelebA data set, we show the traversal of the latent dimensions corresponding to, 'smiling', 'color temperature', 'brightness of background', and 'hair color'. For the 3D Chairs data set, the latent dimensions correspond to 'chair↔arm chair' and 'color', 'wheels', and 'thickness of legs and back'. For the auto-encoder on the 3D Chairs data set, even among a variety of hyper-parameters, training always converges to the plain white image as can be seen in the supplementary material.

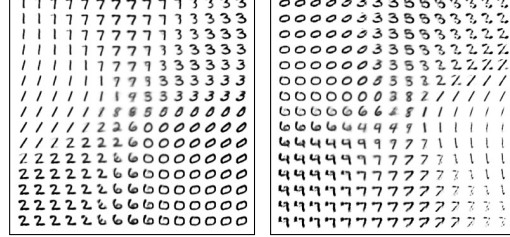

Figure 2: Learned MNIST manifolds. Left: sampling-free VAE. Right: sampling VAE. The latent space is 2 dimensional and the displayed range is $[-5, 5]$. Training details are in the supplementary material.

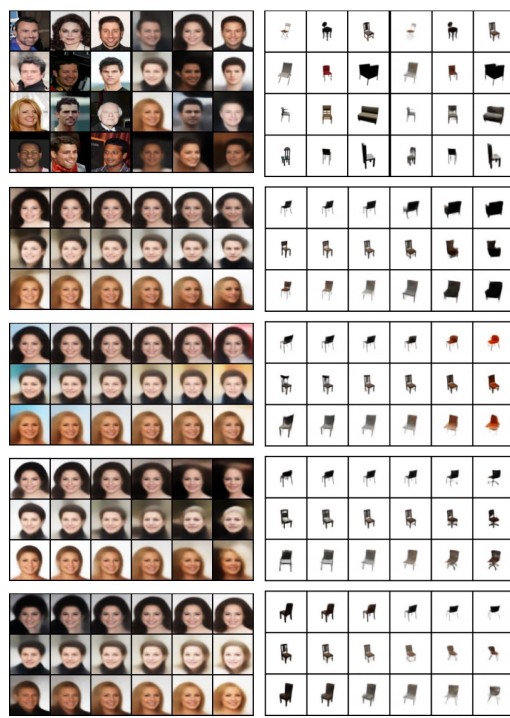

Figure 3: Reconstructions and traversals for the CelebA (left) and 3D Chairs (right) data sets. The traversals are in the range of $[-3, 3]$. An exhaustive display of all latent dimensions as well as a comparison to sampling VAEs and AEs is provided in the supplementary material.

**Quantitative Evaluation** To quantitatively evaluate the sampling-free approximate VAE, we compare the Frechet Inception Distance (FID) as well as the estimated marginal likelihood (ML) between a conventional sampling VAE and a sampling-free VAE as shown in Figure 4. The FID quantifies how close the images produced by the VAE's data generating process are to the original data set as well as the quality of the produced images. For the FID, the distribution of images sampled from the VAE is compared to the distribution of images in the data set by utilizing embeddings produced by an Inception neural network. The ML quantifies the quality of a VAE by quantifying how close

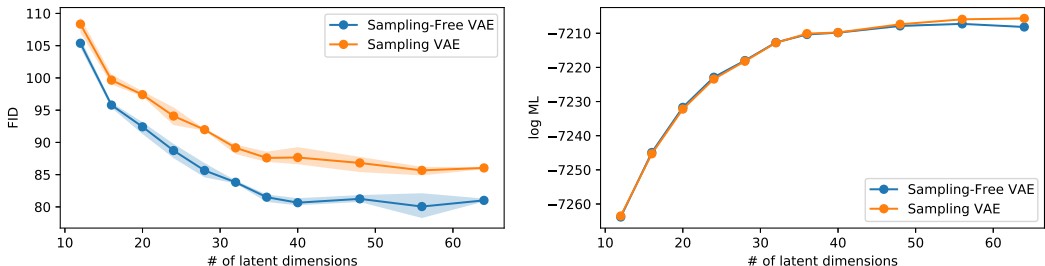

Figure 4: Quantitative evaluation of FID (left) and log marginal likelihood (right) on the CelebA data set for different sizes of the latent space. On FID, the sampling-free Gaussian VAE significantly outperforms the sampling Gaussian VAE and performs on par regarding the log marginal likelihood. Results average over 3 runs and minimum / maximum are marked.

posteriors are to the priors as well as considering the reconstruction error under sampling. For the ML, for each image, the marginal likelihood is estimated via many samples (50 000). We found that for CelebA, for each size of the latent space, the FID of the sampling-free VAE is significantly better than the FID of the sampling VAE. Furthermore, we found that the ML of the sampling-free VAEs is very close to the ML of the sampling VAEs, for some latent dimensionalities, the sampling-free VAE is better, while for others the sampling VAE is better. Plots for the 3D Chairs data set can be found in the supplementary material. We emphasize that we evaluated the sampling-free VAE also via sampling such that there is no bias in the estimates.

To evaluate the models based on computational budget, we additionally report results for the sampling-free VAE to match the computational budget of the sampling VAE's full training. Specifically, we evaluate the sampling-free variant after 650 000 training steps, while the sampling counterpart trains for 1 000 000 steps. The results are shown in Table 2. Even in this setting, the sampling-free formulation achieves a better on FID than the sampling VAE on the CelebA data set, however, the estimated log marginal likelihood suffers.

Table 2: Quantitative evaluation for the sampling-free VAE, where we include the performance of the sampling-free VAE with shorter training such that the total training budget matches the total full training time of the sampling VAE. Results are reported for a 16 dimensional latent space and averaged over 3 runs.

| Method | CelebA | | 3D Chairs | |
|---|---|---|---|---|
| | FID | log ML | FID | log ML |
| Sampling-Free Gaussian VAE | 95.79 | −7244.9 | 51.96 | −7078.8 |
| Sampling Gaussian VAE | 99.66 | −7245.3 | 53.53 | −7078.9 |
| Sampling-Free (matching budget) | 98.00 | −7247.1 | 54.21 | −7079.6 |

### 4.3 COMPARISON TO OTHER GRADIENT ESTIMATE VARIANCE REDUCTION METHODS

We compare our method to the Sticking-the-Landing (Stick) method by Roeder *et al.* [26] and to the importance weighted auto-encoders (IWAE) by Burda *et al.* [23]. For this, we train a VAE on the MNIST [16] and Omniglot [43] data sets and for the sampling methods (vanilla sampling VAE, Stick, IWAE) we vary the number of samples. The encoder and decoder of the architecture each have two hidden layers with tanh activation functions and 200 neurons per layer. We train the model at a batch size of 20 using the Adam optimizer [42]. This follows the setup by Roeder *et al.* [26]. However, a difference is that we model the image space with Gaussians in contrast to the Bernoulli outputs used by Roeder *et al.* [26]; as the method ([26]) readily supports Gaussian outputs, we use it with Gaussian outputs. As tanh is not piece-wise linear, we use the first-order Taylor approximation in this case.

Table 3: Evaluation on MNIST (left) and Omniglot (right) with different methods and numbers of samples. The metric is the NLL on the test set (smaller is better) averaged over 5 runs.

| Samples | Ours | vanilla VAE | Stick | IWAE | Samples | Ours | vanilla VAE | Stick | IWAE |
|---|---|---|---|---|---|---|---|---|---|
| 1 | **26.18** | 27.61 | 26.79 | 27.62 | 1 | **30.14** | 31.49 | 30.99 | 31.49 |
| 5 | — | 26.98 | 26.50 | 26.44 | 5 | — | 31.15 | 30.75 | 30.91 |
| 20 | — | 26.62 | 26.48 | **26.11** | 20 | — | 30.88 | 30.73 | 30.51 |
| 50 | — | 26.64 | 26.59 | **26.08** | 50 | — | 30.83 | 30.67 | 30.36 |
| 100 | — | 26.58 | 26.79 | **25.99** | 100 | — | 30.76 | 30.73 | 30.34 |
| 200 | — | 26.70 | 26.70 | **25.95** | 200 | — | 30.72 | 30.70 | 30.30 |

Table 4: Posterior collapse experiment: VAEs trained on MNIST with a 200-dimensional latent space. Full batch training on a subset of $1\,000$ MNIST images for a controlled setting where for the sampling-free VAE no stochastic effects are present, and for the sampling approach only the stochasticity of the reparameterization is present. Results are averaged over 10 runs.

| Method / Metric | $(\epsilon, \delta)-$posterior collapse | | Reconstruction Error | | KL-Divergence | |
|---|---|---|---|---|---|---|
| Epochs | 100 | 1 000 | 100 | 1 000 | 100 | 1 000 |
| Sampling VAE | **99.0%** $(\pm 1.5\%)$ | **60.0%** $(\pm 15.7\%)$ | 172.5 $(\pm 3.0)$ | 46.2 $(\pm 12.4)$ | 0.1 $(\pm 0.1)$ | 4.8 $(\pm 1.3)$ |
| Sampling-Free VAE | **28.4%** $(\pm 4.2\%)$ | **0.6%** $(\pm 0.6\%)$ | 38.7 $(\pm 0.5)$ | 9.8 $(\pm 0.4)$ | 2.9 $(\pm 0.1)$ | 4.2 $(\pm 0.1)$ |

We report the results in Table 3. We can see that on both data sets, the sampling-free method achieves competitive performance. The sampling VAE and the Sticking-the-Landing method do not achieve the same NLL even with 200 samples. Only the IWAE can outperform the sampling-free method on MNIST with at least 20 samples. On Omniglot, our method also performs better than IWAE with 200 samples. Overall, the results show that the sampling-free approximation performs very well, even when compared to training with large numbers of samples.

## 4.4 POSTERIOR COLLAPSE

We analyze the effect of a sampling-free approximate formulation of deep VAEs on the problem of posterior collapse. For this, we use the notion of an $(\epsilon, \delta)-$collapse of a dimension introduced by Lucas *et al.* [13]. It is defined as follows: A latent dimension $i$ has $(\epsilon, \delta)-$collapsed if

$$\mathbb{E}_{x \sim p}\left(D_{\mathrm{KL}}(q_x(z_i), p(z_i)) < \epsilon\right) \geq 1 - \delta. \tag{22}$$

We follow Lucas *et al.* [13] in setting $\delta = 0.01$. We replicate their setting of training a VAE with 2 hidden layers and 200 latent dimensions for $1\,000$ steps with full batch training on a subset of $1\,000$ MNIST images. We report the fraction of dimensions that have $(\epsilon, \delta)-$collapsed where $\epsilon = 0.05, \delta = 0.01$, the reconstruction error, as well as the KL divergence after 100 and $1\,000$ training steps. We report these results averaged over 10 runs in Table 4.

We find that, for the sampling-free VAE, the reconstruction error is smaller than the reconstruction error for the sampling VAE. While the sampling VAE has a posterior collapse in $60\%$ of the latent dimensions, the sampling-free VAE suffers from posterior collapse in only $0.6\%$ of the latent dimensions. Simultaneously, the KL-Divergence is lower for the sampling-free VAE. These results indicate that the sampling-free VAE is more robust against $(\epsilon, \delta)-$collapse than the sampling VAE.

## 4.5 RUNTIME ANALYSIS

Finally, we discuss the runtime of the proposed approach and compare it to its sampling counterpart. For this analysis, we compare the training times of both methods on the CelebA data set for $10^6$ steps at a batch size of $64$ and 16 latent dimensions on a single Quadro RTX 6000 GPU. For the sampling VAE we measure an average training time of 9.7 hours, while the proposed sampling-free formulation takes on average 14.5 hours on the same GPU. Note that, in case of the sampling-free VAE, the computationally most expensive component is computing the Jacobian. Thus, if the hidden dimension would be very large, the computational overhead would scale linearly ($\mathcal{O}(m)$) with the dimensionality of the hidden space $m$. Note that the overhead of our method is small in comparison to the (in the number of neurons) exponential overhead of computing the exact sampling-free ELBO.

## 5 DISCUSSION & CONCLUSION

We proposed a novel approximate sampling-free approximation of Gaussian VAEs. Our formulation is reasonably fast to compute, i.e., it can be computed in linear time in the number of latent dimensions, contrasting prior art which had an exponential complexity for computing the exact objective. We found that the proposed formulation achieves a significantly lower variance in estimating the objectives than the sampling approach. However, while the sampling approach is unbiased, our lower-variance approximation is not unbiased. Nevertheless, our approximation learns better Gaussian VAEs than its sampling counterpart and can even compete when the computational budget is fixed. Furthermore, we observe that the sampling-free VAE is more robust against posterior collapse than the sampling VAE. We hope to have paved the way for future research to analyze and understand the behavior of VAEs.

REPRODUCIBILITY

In this work, we only use public data sets. We implemented all experiments in PyTorch [44] and our implementation will be made publicly available upon publication. Network architectures and training hyperparameters are specified in the supplementary material.

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

## A    TAYLOR SERIES VISUALIZATION

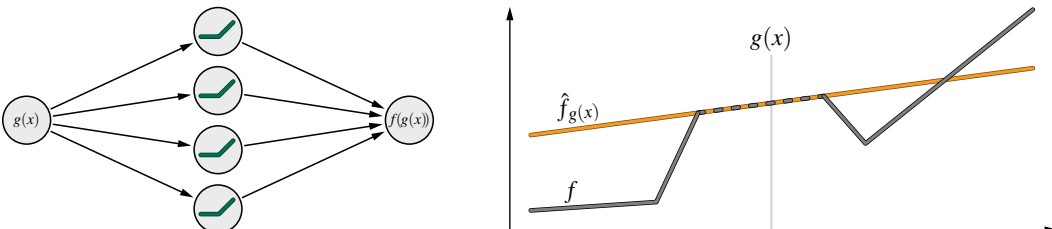

Figure A.1: Example of a Taylor series expansion of a locally affine neural network at point $g(x)$. $f$ (gray) is the original function / neural network, and $\hat{f}_{g(x)}$ (orange) is the Taylor series of $f$ at point $g(x)$. As $g(x)$ is the mean of the latent normal distribution that is transformed by $f$, the Taylor series at point $g(x)$ gives a good approximation in the vicinity of $g(x)$.

## B    IMPLEMENTATION DETAILS

In the following, we discuss implementation details including network architectures.

Throughout the experiments in this work, we use $c = 0.5$, which coincides with many public implementations (which omit the prefactor of $\frac{1}{2c}$ / set this to 1).

### B.1    CELEBA AND 3D CHAIRS

**Network Architecture**    For the CelebA and 3D Chairs experiments, we use the architecture by Higgins *et al.* [3]. The encoder consists of five convolutional layers with kernel size $4 \times 4$, a stride of 2, and (32, 32, 64, 64, 256) latent dimensions with ReLU activations. Following that, a fully-connected layer mapping to the latent space. The explicit architecture is: `Conv 32x4x4 (stride 2), ReLU,` `Conv 32x4x4 (stride 2), ReLU, Conv 64x4x4 (stride 2), ReLU, Conv 64x4x4` `(stride 2), ReLU, Conv 256x4x4 (stride 1), ReLU, FC 256→latent_dim`.

The decoder is the deconvolutional reverse of the encoder. In the qualitative evaluation, we use a latent dimension of 32 for CelebA, and a latent dimension of 16 for 3D Chairs. In the quantitative evaluation, we use the range of $12 - 64$ for CelebA, and the range of $8 - 16$ for 3D Chairs.

**Training**    We trained the CelebA and 3D Chairs models for $10^6$ iterations with the Adam optimizer [42] at a learning rate of $10^{-4}$.

### B.2    MNIST

**Network Architecture**    Here, we use a fully connected network architecture. The encoder consists of three fully connected layers with hidden dimensions (512, 256) and with ReLU activations. The decoder is the reverse of the encoder. For the 2D traversal experiment, we use a latent dimension of 2 and for the experiments examining posterior collapse a latent dimension of 200.

**Training**    We trained the MNIST model for the 2D traversal for $5 \cdot 10^4$ iterations with the Adam optimizer [42] at a learning rate of $10^{-4}$. For the experiments examining posterior collapse, we trained for $10^3$ iterations / epochs at full batch training on a reduced data set of $1\,000$ random samples and with a learning rate of $10^{-3}$.

## C    ADDITIONAL RESULTS

### C.1    QUANTITATIVE EVALUATION FOR 3D CHAIRS

See Figure C.1 for the quantitative performance of the sampling-free VAE compared to a sampling VAE for different latent dimensionalities.

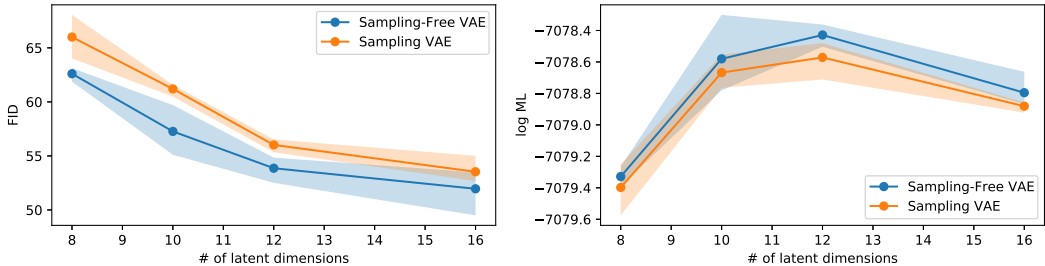

Figure C.1: Quantitative evaluation of FID (left) and log marginal likelihood (right) on the 3D Chairs data set for different sizes of the latent space. On FID (lower is better), the sampling-free VAE significantly outperforms the sampling VAE and performs on par / better regarding the log marginal likelihood (larger is better). Results average over 3 runs and minimum / maximum are marked.

## C.2 STANDARD DEVIATIONS

Table 5: Table 2 but with standard deviations.

| Method | CelebA | | 3D Chairs | |
|---|---|---|---|---|
| | FID | log ML | FID | log ML |
| Sampling-Free Gaussian VAE | $95.79 \pm 0.33$ | $-7244.9 \pm 0.6$ | $51.96 \pm 1.75$ | $-7078.8 \pm 0.1$ |
| Sampling Gaussian VAE | $99.66 \pm 0.65$ | $-7245.3 \pm 0.1$ | $53.53 \pm 1.05$ | $-7078.9 \pm 0.0$ |
| Sampling-Free (matching budget) | $98.00 \pm 1.79$ | $-7247.1 \pm 0.4$ | $54.21 \pm 1.63$ | $-7079.6 \pm 0.1$ |

## C.3 GAP BETWEEN UNBIASED ELBO AND BIASED ELBO

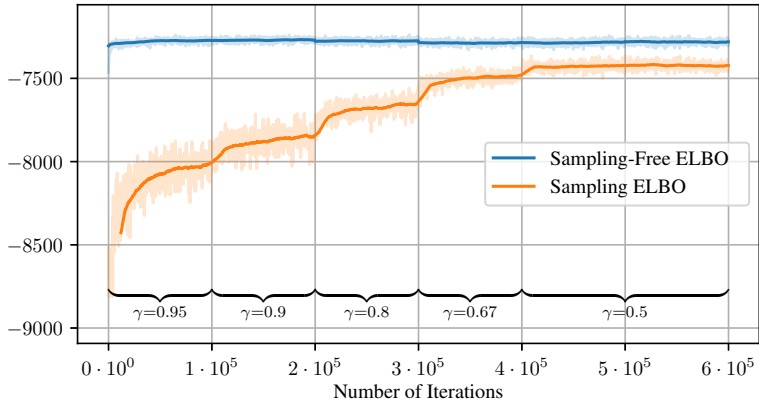

Figure C.2: Gap between unbiased ELBO (Sampling ELBO) and biased ELBO (Sampling-Free ELBO) for different $\gamma$s. Both lines are of the same model, which is trained with the biased ELBO and evaluated with both methods. Same training setting as in Figure 1.

## C.4 FURTHER EXAMPLES, FULL LATENT TRAVERSALS, AND COMPARISON TO OTHER METHODS

In Figures C.3–C.4, we show reconstructions and full latent traversals for 3D Chairs for the sampling-free VAE, the sampling VAE, the sampling $\beta-$VAE, and an auto-encodere. In Figures C.5–C.8, we show the same for CelebA. For the auto-encoder on the 3D Chairs data set, even among a variety of hyper-parameters, training always converges to the plain white image.

Figure C.3: Additional reconstructions and all traversals for the 3D Chairs data set (part 1).

Figure C.4: Additional reconstructions and all traversals for the 3D Chairs data set (part 2).

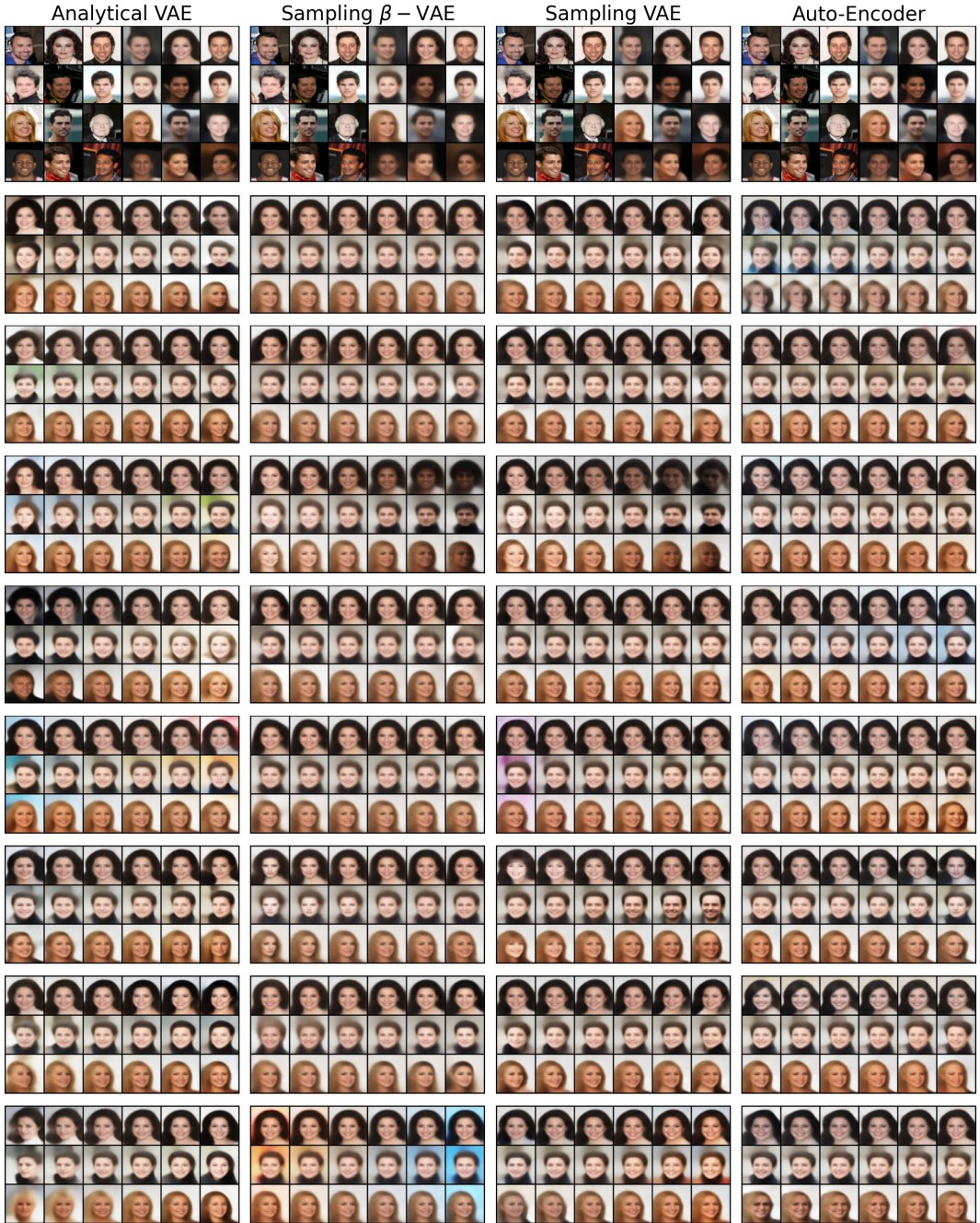

Figure C.5: Reconstructions and traversals for the CelebA data set with 32 latent dimensions (part 1).

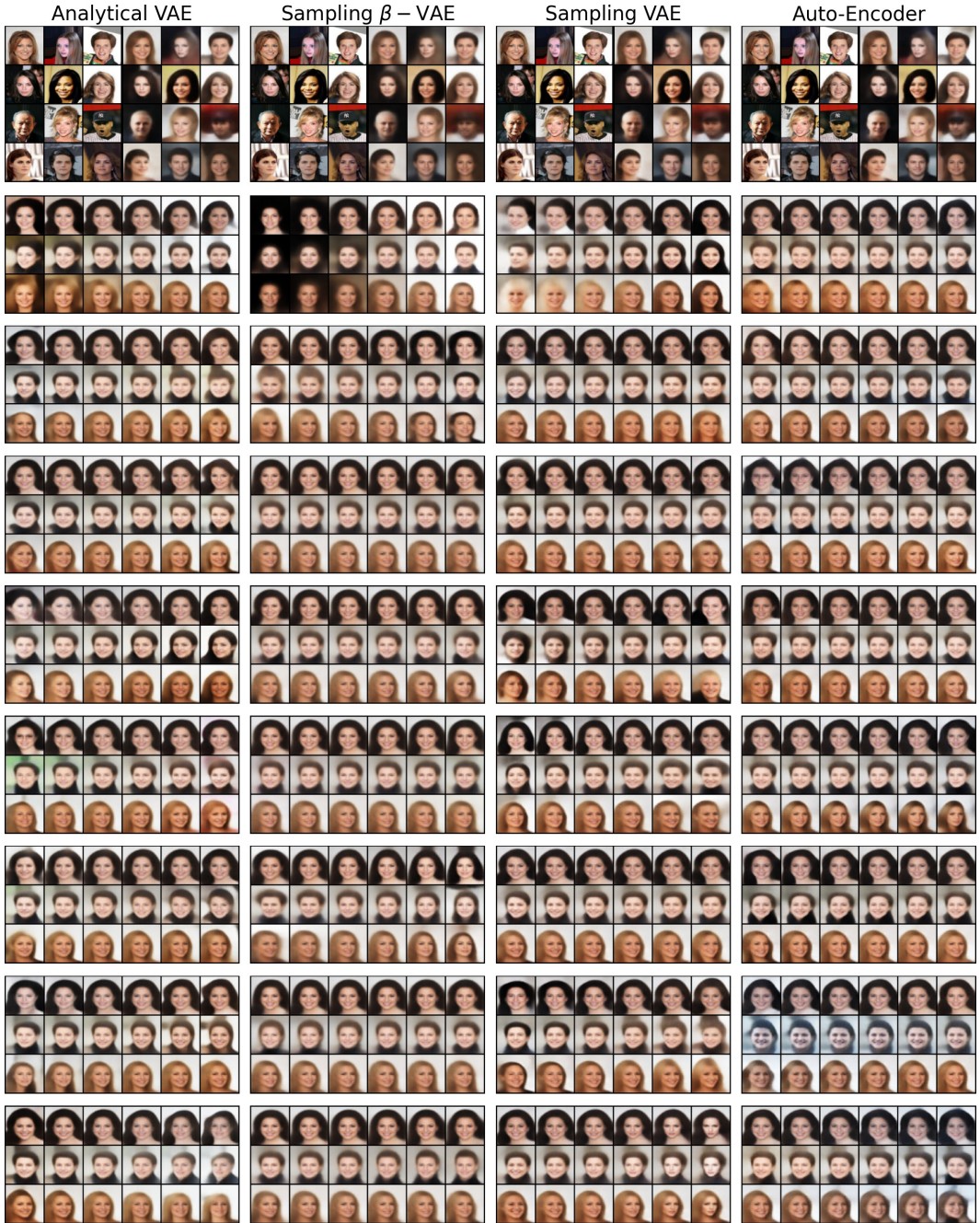

Figure C.6: Reconstructions and traversals for the CelebA data set with 32 latent dimensions (part 2).

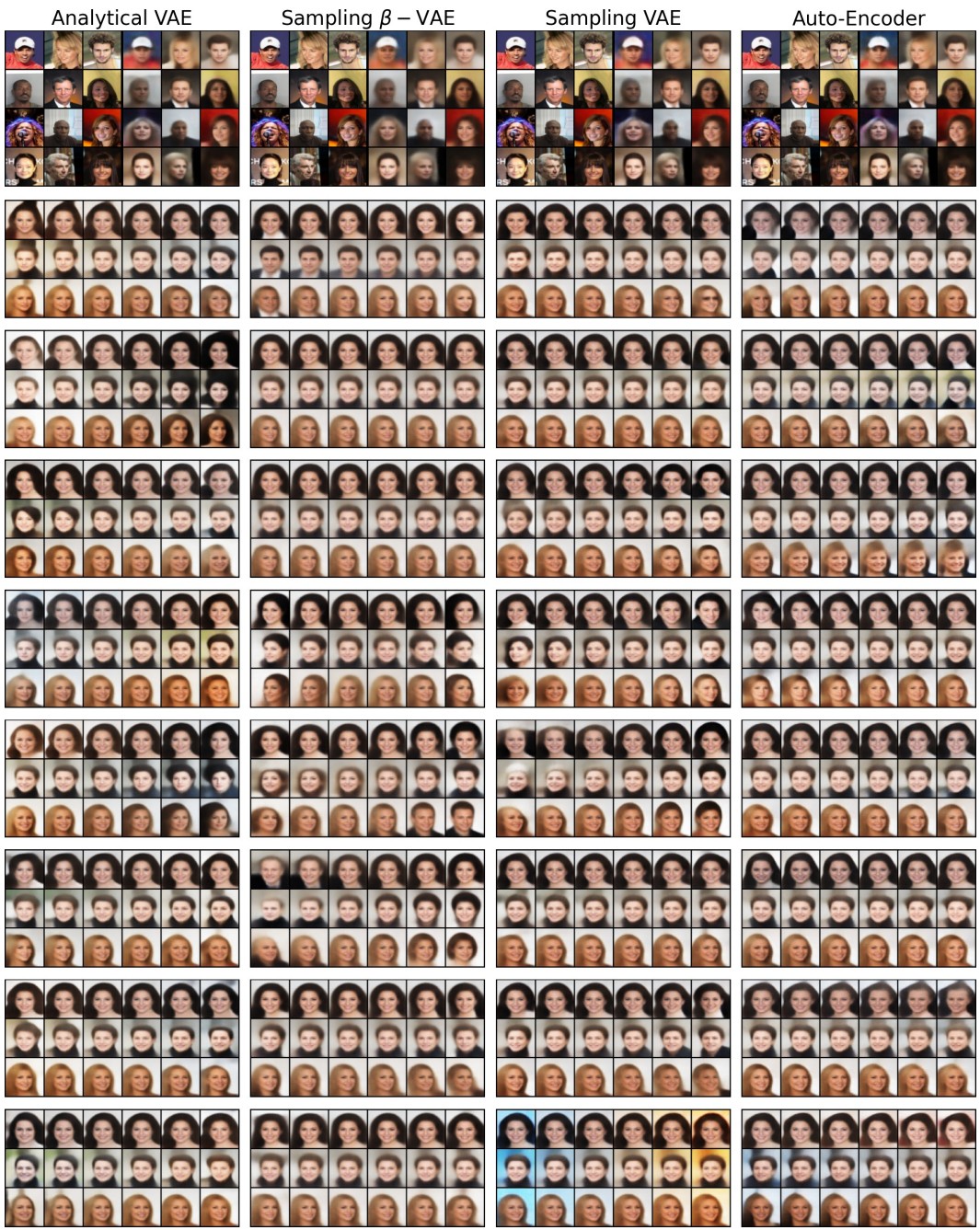

Figure C.7: Reconstructions and traversals for the CelebA data set with 32 latent dimensions (part 3).

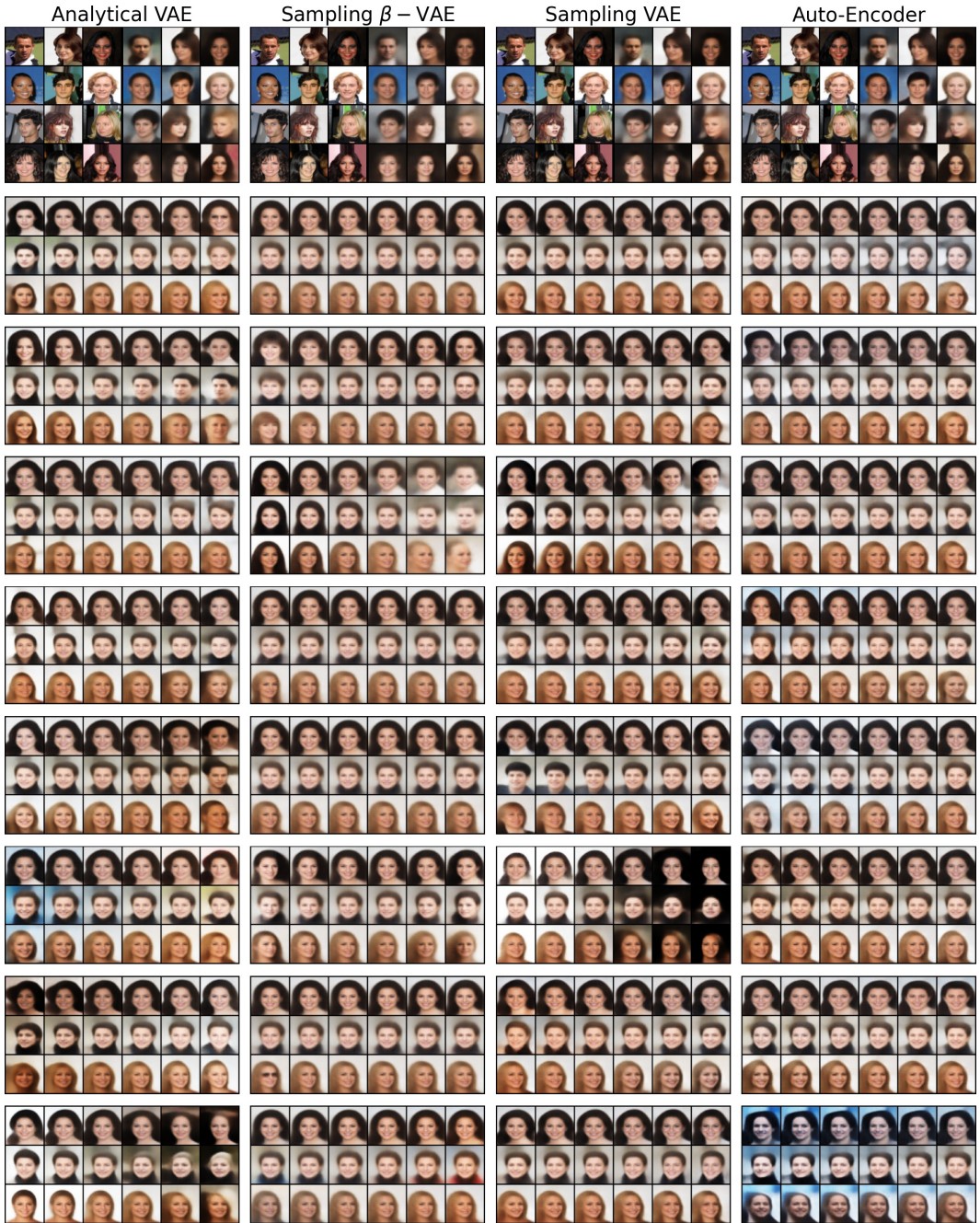

Figure C.8: Reconstructions and traversals for the CelebA data set with 32 latent dimensions (part 4).

