# OpenReview forum: "A Sampling-Free Approximation of Gaussian Variational Auto-Encoders"
_ICLR.cc/2022/Conference — ICLR 2022 Submitted_

### Official Review · Reviewer_smp3 · 2021-11-01

**Correctness:** 4
**Technical Novelty And Significance:** 2
**Empirical Novelty And Significance:** 2
**Recommendation:** 5
**Confidence:** 5

**Main Review:**

_Edit: Thank you for the rebuttal. The authors clarify some of my concerns and include some minor improvements in the experimental comparisons, which is why I slightly increase the score (3 $\to$ 5). However, the overall contribution still remains minor in my opinion which is why I can't recommend acceptance._
__________


### Strengths
- The paper is overall well written, and the approach is evaluated on several data sets.
- The computational cost introduced by the approach is explicitly discussed and not hidden in an appendix.

### Weaknesses
- The contribution itself is very minor, as Taylor approximations to solve expectations are a common approach in the literature. (See, e.g. Wang and Blei (2013); Blei et al. (2017) and the references therein for further pointers, or the keyword delta method.)
- The sampling-free method aims to offer a better approximation to the ELBO. However, the ELBO in itself is only of limited use and mainly serves as a gradient signal. While the experiments seem to show that the approach improves upon a naive sampling, the paper lacks a comparison (and a discussion) to other methods that aim to stabilize the gradient e, e.g. Roeder et al. (2017) and similar methods.
- Similarly, a comparison with other approaches of similar network depths to properly place, e.g. its FID into relation with the current literature, is missing.

### Minor points & Questions
- How to read Figure 1 and the $\gamma$ discussion? Sec 3.3 first mentions the exploration of three $\gamma$ values (0.5,0.9,0.99), while Figure 1 explores a different set of values, which influence only the orange sampling ELBO, not the blue sampling-free ELBO. Is the legend broken here? Similarly, what does the number of iterations refer to, as the blue line seems to have converged from the beginning?
- Figures 7,8 in the appendix seem to be broken for the Autoencoder. Is that an error in the figure or due to problems with the underlying model?
- Sec 4.1 mentions the "true output... via a large number of samples". What is large here? _(I might have overlooked the hyperparameter in the appendix)_
- What is the precise architecture used in the experiments? Sec 4.2 first claims to follow Burgess et al.'s architecture, then later mentions a 5 conv layers + 1 fully connected layer structure. Appendix B.1 switches to claiming to follow Higgins et al. instead but mentions again five convolutional layers, which is not what Higgins et al. report to be using in their appendix (they claim four convolutional layers). (Burgess et al. also use four convolutional layers but a different number of channels compared to Higgins.)
- Are the minimum/maximum missing from the right plot in Figure 4, or so close to the mean to not be visible?
- Table 2 gives the average over multiple runs. What is the standard deviation for the results?


__________
Wang and Blei, Variational Inference in Nonconjugate Models, JMLR 2013
Blei et al., Variational Inference: A Review for Statisticians, JASA, 2017
Roeder et al., Sticking the landing: Simple, lower-variance gradient estimators for variational inference, NeuRIPS 2017


**Summary Of The Paper:**

The paper proposes a sampling-free approximation to the ELBO of a variational auto-encoder with Gaussian likelihood and mean-field Gaussian variational posterior. The approach relies on a Taylor series around the mean of the posterior, which allows for an evaluation of the ELBO's expectation, rather than relying on Monte Carlo Sampling.


**Summary Of The Review:**

The paper demonstrates the usefulness of a Taylor approximation compared to a sampling-based approach. However, that in itself is of limited novelty, and the paper lacks a proper comparison to other methods and the broader literature to classify it properly.

---

> ### Author Response · Authors · 2021-11-22
> **Response to Reviewer smp3 (3/3)**
>
> > Figures 7,8 in the appendix seem to be broken for the Autoencoder. Is that an error in the figure or due to problems with the underlying model?
>
> This is a problem with the underlying model / data set.
> We also tried other hyperparameters, but the AE always converges to the white background.
> This is because the conventional auto-encoder gets stuck in the local minimum of predicting a white plane, which is actually a relatively good model for its metric considering that the background covers most of the image.
> This is also described in Section 4.2 in the second paragraph of "Qualitative Evaluation". We clarified this locally in the revision by including the information in the supplementary material.
>
> > Sec 4.1 mentions the "true output... via a large number of samples". What is large here? (I might have overlooked the hyperparameter in the appendix)
>
> It is $15\,000$ samples. We included this detail in the revision.
>
> > What is the precise architecture used in the experiments? Sec 4.2 first claims to follow Burgess et al.'s architecture, then later mentions a 5 conv layers + 1 fully connected layer structure. Appendix B.1 switches to claiming to follow Higgins et al. instead but mentions again five convolutional layers, which is not what Higgins et al. report to be using in their appendix (they claim four convolutional layers). (Burgess et al. also use four convolutional layers but a different number of channels compared to Higgins.)
>
> We apologize for the confusion. We use the model by Higgins et al.
> The Higgins et al. architecture may either be implemented as 5 conv layers + 1 fully connected layer or as 4 conv layers + 2 fully connected layers. Higgins et al. describe it as the latter one.
> The equivalence is because after the first 4 conv layers, the tensor has a shape of 64x4x4, and applying a conv layer with a kernel of 4x4 and 256 channels returns a tensor of shape 256x1x1 and this is equivalent to applying a respective fully connected layer to the flattened vector of 64x4x4.
> We explicitly write our encoder model architecture as
>
> Conv 32x4x4 (stride 2), ReLU, Conv 32x4x4 (stride 2), ReLU, Conv 64x4x4 (stride 2), ReLU, Conv 64x4x4 (stride 2), ReLU, Conv 256x4x4 (stride 1), ReLU, FC 256$\to$latent_dim
>
> and the decoder as its reversed ConvTranspose counter-part.
>
> > Are the minimum/maximum missing from the right plot in Figure 4, or so close to the mean to not be visible?
>
> They are so close to the mean that they are not visible.
>
> > Table 2 gives the average over multiple runs. What is the standard deviation for the results?
>
> See the following table.
>
> | Method | FID (CelebA) | log ML (CelebA)| FID (3D Chairs) | log ML (3D Chairs) |
> |--------|--------------|----------------|-----------------|--------------------|
> | Sampling-Free Gaussian VAE | $95.79 \pm 0.33$ | $-7244.9 \pm 0.6$ | $51.96 \pm 1.75$ | $-7078.8 \pm 0.1$ |
> | Sampling Gaussian VAE | $99.66 \pm 0.65$ | $-7245.3 \pm 0.1$ | $53.53 \pm 1.05$ | $-7078.9 \pm 0.0$ |
> | Sampling-Free (matching budget) | $98.00 \pm 1.79$ | $-7247.1 \pm 0.4$ | $54.21 \pm 1.63$ | $-7079.6 \pm 0.1$ |
>
> We included the table with standard deviations in the supplementary material of the revision.

---

> ### Author Response · Authors · 2021-11-22
> **Response to Reviewer smp3 (2/3)**
>
> > How to read Figure 1 and the discussion? Sec 3.3 first mentions the exploration of three values (0.5,0.9,0.99), while Figure 1 explores a different set of values, which influence only the orange sampling ELBO, not the blue sampling-free ELBO. Is the legend broken here? Similarly, what does the number of iterations refer to, as the blue line seems to have converged from the beginning?
>
> We apologize for the unclear presentation of Section 3.3. Therefore, we rewrote Section 3.3 to clarify it.
>
> To answer your specific questions:
>
> In general, setting $\gamma=1$ usually does not allow training or makes the training diverge due to numerical problems, therefore $\gamma$ should be smaller than or equal to $0.99$ (the exact number depends on many factors such as data set, model, optimizer, learning rate, and float32/float64 data types).
> However, there is another aspect to $\gamma$, which is reducing the bias of the ELBO approximation.
> Empirically, we found that tuning $\gamma$ (which can be seen as a kind of Tikhonov regularization) can drastically reduce the bias of the estimate to make it more aligned with the true objective.
> In the old Figure 1, we demonstrated this behavior in the following way: we train a sampling-free VAE and measure the biased sampling-free ELBO estimate (blue) as well as the unbiased sampled ELBO estimate (orange).
> We divide the training into 5 regions and use a different $\gamma$ in each region.
> We observe that the biased ELBO estimate is always above the unbiased estimate. This depends on data set, model, etc. and is just the bias of the estimator, and this does not matter much as a constant is irrelevant when we are interested in gradients for training. The fact that the biased sampling-free ELBO approximation did not change "much" (in a visual sense) was just due to the displayed scale.
>
> The overall goal is increasing the unbiased sampling ELBO.
> With $\gamma=0.95$, the training converges from -8750 to -8000. With $\gamma=0.9$, the training converges to -7800, and so on... With $\gamma=0.5$, it converges to around -7450.
> The goal of the plot is to demonstrate that by choosing $\gamma$, we can reduce the bias of our approximation and the trained model achieves a better (true and unbiased) ELBO.
>
> The number of iterations refers to the number of training steps.

---

> ### Author Response · Authors · 2021-11-22
> **Response to Reviewer smp3 (1/3)**
>
> Thanks a lot for taking the time to carefully review our paper!
> We especially appreciate the additional references that you provide, which significantly improves the presentation and comparison to related work.
> We have conducted new experiments and, following your suggestion, we now compare ourselves to Sticking-the-Landing by Roeder et al. (2017) and IWAE by Burda et al. (2016).
> We hope our responses address all of your concerns.
>
> > The sampling-free method aims to offer a better approximation to the ELBO. However, the ELBO in itself is only of limited use and mainly serves as a gradient signal. While the experiments seem to show that the approach improves upon a naive sampling, the paper lacks a comparison (and a discussion) to other methods that aim to stabilize the gradient e, e.g. Roeder et al. (2017) and similar methods.
>
> The aim of our sampling-free method actually is to obtain better gradient signals to train the VAE most effectively, and is *not* to achieve a better approximation to the ELBO.
> This is also what the experiments show: while the unbiased ELBO estimate is not accurate, it has no variance (because it does not sample) and using it as a loss yields good gradients that improve the unbiased ELBO (/ NLL / FID).
>
> Therefore, a comparison to other methods stabilizing the gradient is indeed very relevant; thus, thank you for pointing us to Roeder et al. (2017).
>
> Correspondingly, we have conducted new experiments on the MNIST and Omniglot data sets, where we compare our method to the Sticking-the-Landing and Importance Weighted Autoencoder methods.
> We describe these experiments and its results above in the comment `New Experiments`.
>
> > The contribution itself is very minor, as Taylor approximations to solve expectations are a common approach in the literature. (See, e.g. Wang and Blei (2013); Blei et al. (2017) and the references therein for further pointers, or the keyword delta method.)
>
> We agree that Taylor approximations are ubiquitous and are also used to solve expectations in the literature.
> However, we do not believe that this should be considered a weakness, as using Taylor approximations by themselves is a common practice, and the novelty lies in the approximation of the training objective (and not in the Taylor series approximation, which is only a method we use within our approximation).
>
> VI methods, such as the delta method or the Laplace method (as it is also described in Wang and Blei (2013)), require solving an optimization problem to obtain an approximation of the output distribution.
> This means that for training the VAE, the optimization problem would have to be solved for each sample and each training step.
> Thus, training a VAE but using the delta method would be an optimization problem within an optimization problem, and therefore substantially more expensive than our method.
> In contrast, our method uses the Taylor approximation to produce a closed-form estimate of the training objective and the only optimization is the training of the VAE.
>
> We included the references and a discussion of the delta method and the Laplace method in the related work section.
>
> > Similarly, a comparison with other approaches of similar network depths to properly place, e.g. its FID into relation with the current literature, is missing.
>
> We propose to add this to the final revision of the paper.
> Adding it to the response was not possible in the short time period.

---

### Official Review · Reviewer_Zwr6 · 2021-11-01

**Correctness:** 3
**Technical Novelty And Significance:** 2
**Empirical Novelty And Significance:** 2
**Recommendation:** 5
**Confidence:** 5

**Main Review:**

Post-rebuttal: Thank you for the rebuttal. The authors addressed some of my concerns, added experimental comparison and updated the text. However, the overall novelty and contributions are still not adequate for ICLR in my opinion. I will not change the score.

######

Pros:

+ The analytical approximation would help with reducing the variance and computational expense of the ELBO.

+ The proposed method achieves similar or better performance than its sampling counterpart.

+ The proposed method is found alleviated the issue of posterior collapse that the sampling VAEs suffers.


Cons:

- The writing should be substantially improved. The words and sentences are not so fluent. There are obvious grammar errors and typos. I believe that it is critical for the readers to correctly understand and evaluate the contributions. The figures are labeled in a random order.

- The advantages in performance looks marginal comparing to the sampling VAE. The proposed method beats sampling VAE for small number of samples. I don't mean to criticize this point since it's an approximation while the sampling ELBO is an unbiased estimate. Instead, the claimed advantages are not well demonstrated. For example, the authors could show the wall time and variance of ELBO. Sec 4.4 is qualitative.

- The proposed approximation is limited to Gaussian encoders and ReLU decoder networks. The Taylor expansion itself should be applicable to other types of networks, but the performance then needs further evaluations.

- Posterior collapse part could be more elaborated. Can the authors address more how this approximation could solve the issue?

Concerns:

* What's its connection to the Laplace method in variational inference.
* Strictly speaking, the approximation is not the exact ELBO using in the sampling VAE and thus different objective in fact. Futhermore, it's not necessarily a lower bound to the marginal likelihood. What is the gap between the approximation and the ELBO or true marginal likelihood?
* The covariance of reconstructions evaluated in the paper is not clearly defined? For approximate VAE and sampling VAE, the output is different. The former is a distribution but the latter is a sample.
* Sec 3.2: Why is the intractability due to exponential complexity rather than integrability?

Minor:

* The computional complexity better be in form of big O notion.
* The layout of Eq. 16 better be topdown.
* Figure 5 ref before Figure 1

**Summary Of The Paper:**

The paper proposes an approximation that frees the calculation of ELBOs of Gaussian variational autoencoders from sampling. To achieve this, it utilizes Taylor expansion on the decoder networks. The proposed method was evaluated on three typical datasets. The authors also found that the approximation alleviated the issue of posterior collapse to the sampling VAEs.

**Summary Of The Review:**

Overall, I vote for rejecting. I like the idea that analytical approximate ELBO would help the computation. The finding on posterior collapse is also interesting. However, the method and evaluation are not adequately demonstrated. The writing can be largely improved too.

---

> ### Author Response · Authors · 2021-11-22
> **Response to Reviewer Zwr6**
>
> We thank you very much for your review.
> Below, we respond to your concerns and questions.
>
> > The writing should be substantially improved. The words and sentences are not so fluent. There are obvious grammar errors and typos. I believe that it is critical for the readers to correctly understand and evaluate the contributions. The figures are labeled in a random order.
>
> We apologize for possible typos. Could you please point us to any grammar errors and typos that you found? Also, could you please elaborate which words and sentences are not fluent?
> Could you clarify which figure is mislabeled? We checked multiple times but did not find any mislabelings.
>
> > The advantages in performance looks marginal comparing to the sampling VAE. The proposed method beats sampling VAE for small number of samples. I don't mean to criticize this point since it's an approximation while the sampling ELBO is an unbiased estimate. Instead, the claimed advantages are not well demonstrated. For example, the authors could show the wall time and variance of ELBO. Sec 4.4 is qualitative.
>
> We added additional experiments comparing our method against two alternatives that also improve the gradient estimates, as described in a separate general response above.
> Here, we also find that our method outperforms the regular sampling VAE even with 200 samples.
>
> We show the runtime of our method in Section 4.4 using wall time. We are unsure what "Sec 4.4 is qualitative." refers to. Could you please elaborate?
>
> As we compute the approximation to the ELBO sampling-free, the estimator accordingly does not have a variance.
>
> > The proposed approximation is limited to Gaussian encoders and ReLU decoder networks. The Taylor expansion itself should be applicable to other types of networks, but the performance then needs further evaluations.
>
> As Gaussian VAEs are the de-facto standard VAEs in practice, this should not be a problem in most cases.
>
> To evaluate the performance on other types of networks, we now conducted the additional experiments using tanh-activated neural networks, as described in a separate general response above.
> For this, we use the first-order Taylor approximation. The results demonstrate that our method also works with tanh.
>
> > Posterior collapse part could be more elaborated. Can the authors address more how this approximation could solve the issue?
>
> Posterior collapse is still an open problem. The results indicate that sampling might be a source of posterior collapse.
> Posterior collapse could (at least partially) be a variance problem, that is, that due to sampling, there are sometimes gradients that let a dimension collapse and once a dimension is collapsed, it cannot be "expanded".
>
> > What's its connection to the Laplace method in variational inference.
>
> The Laplace method can also use a Taylor approximation.
> However, in contrast to our work, the goal of the Laplace method is to find a fitting posterior distribution by solving an optimization problem (i.e., finding the mode of the posterior).
> This could in theory also be used in conjunction with our proposed method but would mean that we would have to solve an optimization problem (Laplace method) inside each step of an optimization problem (training VAE), which would be computationally very expensive and might also be non-trivial regarding backpropagation.
>
> > Strictly speaking, the approximation is not the exact ELBO using in the sampling VAE and thus different objective in fact. Futhermore, it's not necessarily a lower bound to the marginal likelihood. What is the gap between the approximation and the ELBO or true marginal likelihood?
>
> Correct. The gap was displayed for different $\gamma$ in Figure 1 (which we now moved to the supplementary material as Figure C.2).
> We found that reducing $\gamma$ decreases the gap, i.e., the orange and blue line got close to each other when reducing $\gamma$.
>
> > The covariance of reconstructions evaluated in the paper is not clearly defined? For approximate VAE and sampling VAE, the output is different. The former is a distribution but the latter is a sample.
>
> The covariance produced by our approximation is the $J h(x) J^\top$.
> The covariance for the sampling method is the covariance of a number of samples.
>
> > Sec 3.2: Why is the intractability due to exponential complexity rather than integrability?
>
> The integration has to be done over an exponential number of regions, so the integration is actually what causes the exponential complexity.
>
> We address the minor comments in the revision.

---

### Official Review · Reviewer_AwHy · 2021-11-02

**Correctness:** 4
**Technical Novelty And Significance:** 2
**Empirical Novelty And Significance:** 2
**Recommendation:** 5
**Confidence:** 4

**Main Review:**



Strengths

1. The closed-form approximation reduces the computation cost for computing the ELBO of exact Gaussian VAE.
2. The paper is well organized and clearly written.


Weakness.

1. The proposed method is similar to the well-known moment matching (MM) methods. However,  a discussion of the relationship between the proposed method and MM is missing. In addition, it is better to include a comparison with MM to justify the benefit of the proposed method.

2. The proposed method is limited to Gaussian VAE.  It may not apply to VAE with other distribution.

3.  In experiment 4.1,  how to set the hyper-parameter $\gamma$ in Eq.(21).  It is better to set $\gamma=1$ to justify the theoretical approximation quality.

**Summary Of The Paper:**


In this paper, the authors propose a closed-form approximation for the Gaussian decoder distribution in VAE.  The closed-form is achieved by matching the mean and covariance matrix with the Taylor expansion of the nonlinear mapping.

**Summary Of The Review:**


Overall,  I think the proposed method is similar to the moment matching method. It is marginal novel and limited to the Gaussian case. Thus, in my opinion,  this paper is marginally below the acceptance threshold.

---

> ### Author Response · Authors · 2021-11-22
> **Response to Reviewer AwHy**
>
> Thank you very much for your review!
> We hope that our response clarifies all of your questions and concerns.
>
> > The proposed method is similar to the well-known moment matching (MM) methods. However, a discussion of the relationship between the proposed method and MM is missing. In addition, it is better to include a comparison with MM to justify the benefit of the proposed method.
>
> Agreed. We attempted to use moment matching but found that for deeper networks (5 ReLUs), the standard deviations of the output are off by a factor of empirically around 100-500.
> This is because moment matching does not allow modeling correlations in closed form, so at each ReLU, all correlations have to be discarded.
> This causes the estimates to have a too large bias, and the model does not train in the correct direction.
> To our knowledge, an equivalent to our work but with moment matching has not been proposed.
> If you know any moment matching methods for sampling-free VAEs, could you please include the reference and point us to respective code resources to allow for reproducibility?
>
> > The proposed method is limited to Gaussian VAE. It may not apply to VAE with other distribution.
>
> Yes, this is correct. We also note that Gaussian VAEs are the de-facto standard VAEs in practice.
>
> > In experiment 4.1, how to set the hyper-parameter $\gamma$ in Eq.(21). It is better to set $\gamma=1$ to justify the theoretical approximation quality.
>
> We find that setting it to 1 is not stable for numerical reasons, this can be solved by setting it to 0.99. Furthermore, we found empirically that reducing $\gamma$, e.g., to 0.5 reduces the approximation bias, which improves the performance of the method.

---

### Official Review · Reviewer_SKjV · 2021-11-02

**Correctness:** 2
**Technical Novelty And Significance:** 4
**Empirical Novelty And Significance:** Not applicable
**Recommendation:** 8
**Confidence:** 3

**Main Review:**

*Update after rebuttal* I have read the authors’ response and other reviews. I would like to thank the authors for their response. They have addressed my concerns - limited experiments and poor presentation. Although, the related work section could still use some work in terms of writing (for example, to avoid presentation in a form of a list “A did this, B did that”, but it has been improved. I am therefore increasing my score.

————————————————————

Strong points:
1. Simple yet novel approach for approximating ELBO that allows analytical solution
2. Sampling-free training of Gaussian VAEs that scales to large networks
3. Rather thorough experiments in terms of covering the question from different perspectives, i.e. assessment of Taylor approximation, quantitative performance, posterior collapse discussion

Weak points:
1. Limited experiments in terms of considering other baselines. Related works cover a lot of different VAEs and their training methods, yet almost all the experiments are on contrasting with sampling VAE (without the proper specification which specific sampling VAE is used)
2. Motivation for the sampling free training for VAE in general as a concept is not very well provided considering that it comes with the computational overhead. The text implicitly assumes that sampling free training is obviously better and only additionally support it with better performance
3. Some writing issues. The related work section in particular could see a lot of improvement. Considering the context of the paper, one would expect the related work section to focus on the discussion of the training methods for VAEs and on contrasting sampling vs sampling free methods. Instead a reader finds a generic overview of different VAEs, with some works that do not seem to be really related to the submission apart from being devoted to VAEs, e.g. Tolstikhin et al, or Rezende et al.

I am voting for a weak acceptance of the paper as I found the general idea of the paper to be appealing and worth attention of the community, however the above mentioned weaknesses stop me from higher score. I believe the paper can be made very strong with some work as thorough empirical support and better focussed presentation.

Specific comments/suggestions (not necessarily important for assessment, but points to improve the paper):
1. Mix of reference styles is used with Author and [Number] mixed
2. Related work, first paragraph. 3 layer network, specific number of hidden units and latent dimension, MNIST dataset - these may be too much details for related work description.
3. Related work, last sentence - Not very clear what this is for in this section
4. f_{\theta} is not defined
5. I_m is not defined
6. Figure 1. Not clear what sampling and sampling-free mean here as it seems that sampling one should not depend on \gamma, a hyperparameter of the proposed sampling-free method
7. Figure 4. Why there is no shaded area for log ML plot?
8. It would be interesting to see a comparison between the proposed approximated sampling free method and the exact sampling free method, even though it would mean to rely on a small scale experiment

Minor:
1. Section 3.3. "that works independentLY"



**Summary Of The Paper:**

The paper proposes a learning method for Gaussian variational auto-encoders based on Taylor expansion approximation. This allows an analytical formula for ELBO and therefore avoid the requirement of sampling during training.

**Summary Of The Review:**

The paper seems to provide an interesting idea for training a VAE that allows an analytical solution and therefore does not need sampling, but the empirical evaluation of the method is somewhat limited and presentation of the paper could be significantly improved. Therefore, I am voting for the weak acceptance as I believe the idea is worth to be known by the community but the existing drawbacks stop me from giving a higher score.

---

> ### Author Response · Authors · 2021-11-22
> **Response to Reviewer SKjV**
>
> Thanks a lot for taking the time to carefully review our paper and for your valuable suggestions!
> We hope that our response addresses all of your concerns.
>
> > Limited experiments in terms of considering other baselines. Related works cover a lot of different VAEs and their training methods, yet almost all the experiments are on contrasting with sampling VAE (without the proper specification which specific sampling VAE is used)
>
> We added additional experiments comparing our method against two alternatives that also improve the gradient estimates, as described in a separate general response above.
> For the sampling VAE in our original experiments, we used the vanilla VAE with an ELBO as described in equation (8).
>
> > Some writing issues. The related work section in particular could see a lot of improvement. Considering the context of the paper, one would expect the related work section to focus on the discussion of the training methods for VAEs and on contrasting sampling vs sampling free methods. Instead a reader finds a generic overview of different VAEs, with some works that do not seem to be really related to the submission apart from being devoted to VAEs, e.g. Tolstikhin et al, or Rezende et al.
>
> We reworked the related work section and will polish it for the final paper.
>
> >  I believe the paper can be made very strong with some work as thorough empirical support and better focussed presentation.
>
> We conducted additional experiments comparing it to two variance-reducing methods for sampling VAEs, as described in the separate general response above.
>
> > Mix of reference styles is used with Author and [Number] mixed
>
> Could you please clarify, which specific reference you are referring to? When citing implicitly, we use [number], and when citing explicitly, we use Author et al. [number]. Adding the author name for explicit citations makes sure each sentence has a proper subject (avoiding sentences like "[1] propose a ...").
>
> > Related work, first paragraph. 3 layer network, specific number of hidden units and latent dimension, MNIST dataset - these may be too much details for related work description.
>
> We wanted to emphasize that the network is really small and the most objective way to do so is just stating the size. We propose to change it to "a neural network with a total of 24 neurons".
>
> > Related work, last sentence - Not very clear what this is for in this section
>
> Agreed, the sentence was not necessary. We omit it.
>
> > f_{\theta} is not defined
>
> Thanks for noticing. $f_{\theta}$ is the decoder model. We defined it in the revision.
>
> > I_m is not defined
>
> Thanks for noticing. $I_m$ is the identity matrix of size $m\times m$. We defined it in the revision.
>
> > Figure 1. Not clear what sampling and sampling-free mean here as it seems that sampling one should not depend on \gamma, a hyperparameter of the proposed sampling-free method
>
> Agreed, this was not intuitive.
> Thus, we replaced the figure by a figure that simply displays the unbiased ELBO when training with the sampling-free formulation.
>
> > Figure 4. Why is there no shaded area for log ML plot?
>
> It is just not visible because the minimum/maximum are very close to the mean.
>
> > It would be interesting to see a comparison between the proposed approximated sampling free method and the exact sampling free method, even though it would mean to rely on a small scale experiment
>
> We agree that this would be interesting.
> However, it is questionable how representative results on such a small scale would actually be.
> Recall, Balestriero et al. used a latent dimensionality of 1 and only 8 + 16 neurons.
> So on the one hand this means that matrices will be $1\times 1$ and thus scalars, and on the other hand, this means the output distribution is not even approximately Gaussian as the law of large numbers with the central limit theorem do not apply with 8 + 16 hidden neurons.
>
>
> We fixed the typo.

---

### Author Response · Authors · 2021-11-22
**New Experiments**

To accommodate the concerns regarding the experiments, we added an additional set of experiments, which covers the following aspects:

* A comparison to other methods that aim to stabilize the gradient (as suggested by reviewer smp3)
* A comparison to more baselines for stronger empirical support (as suggested by reviewer SKjV)
* An evaluation with larger numbers of samples for the sampling baselines (as suggested by reviewer Zwr6)
* An evaluation with non-ReLU networks (as suggested by reviewer Zwr6)

Correspondingly, we have conducted new experiments on the MNIST and Omniglot data sets, where we compare our method to the vanilla sampling VAE, to Sticking-the-Landing, and to Importance Weighted Autoencoder methods.
Note that in contrast to Roeder et al. (2017) we use a Gaussian output, which contrasts the Bernoulli outputs used in Roeder et al. (2017).
(This yields different negative log likelihoods than Bernoulli in Roeder et al. (2017).)
We follow Roeder et al. (2017) in using a tanh-activated neural network and use the first-order Taylor approximation in this case.
Below, we report the test NLL (evaluated by drawing 5000 samples for each image of the test set) for each method and for both data sets. The results are averaged over 5 runs.

**MNIST**:

| Num. Samples | Sampling-Free VAE | Sampling VAE | Sticking-the-Landing | IWAE |
|--------------|------|--------------|-------------------|------|
|   1 | 26.18 | 27.61 | 26.79 | 27.62 |
|   5 | ---   | 26.98 | 26.50 | 26.44 |
|  20 | ---   | 26.62 | 26.48 | 26.11 |
|  50 | ---   | 26.64 | 26.59 | 26.08 |
| 100 | ---   | 26.58 | 26.79 | 25.99 |
| 200 | ---   | 26.70 | 26.70 | 25.95 |

**Omniglot**:

| Num. Samples | Sampling-Free VAE | Sampling VAE | Sticking-the-Landing | IWAE |
|--------------|------|--------------|-------------------|------|
|   1 | 30.14 | 31.49 | 30.99 | 31.49 |
|   5 | ---   | 31.15 | 30.75 | 30.91 |
|  20 | ---   | 30.88 | 30.73 | 30.51 |
|  50 | ---   | 30.83 | 30.67 | 30.36 |
| 100 | ---   | 30.76 | 30.73 | 30.34 |
| 200 | ---   | 30.72 | 30.70 | 30.30 |

We can see that in both cases, the sampling-free method achieves competitive performance.
On both data sets, the sampling VAE and the Sticking-the-Landing method do not achieve the same accuracy even with 200 samples.
Only the IWAE can outperform the sampling-free method on MNIST with at least 20 samples.
On Omniglot, our method also performs better than IWAE with 200 samples.

We added these experiments as Section 4.3 to the revision.

---

### Author Response · Authors · 2021-11-22
**Revision**

Dear reviewers and AC,

Thank you all for your time investment and for your comments and constructive criticisms.
Based on the reviewer feedback, we have extended the related work and conducted an additional set of experiments covering multiple new aspects of the evaluation in Section 4.3.
In addition, we rewrote Section 3.3 to clarify the regularization.
We address the individual reviewer concerns and questions in the respective responses and also incorporate it in the revision.

Please don't hesitate to ask if anything is unclear or new questions come up.

Best regards

Authors

---

### Decision · Program_Chairs · 2022-01-20

**Decision:**

Reject

**Comment:**

This paper proposes a way to train Gaussian variational autoencoders that does not require the computation of empirical expectations but instead approximates the decoder network by its Taylor series. Results on 3 datasets show the competitiveness of the approach.

Based on the limited novelty of the approach, three (out of 4) knowledgeable reviewers recommend rejection and I agree. Variational autoencoders are simply doing variational inference in a specific model and, as one of the reviewers has pointed out, these types of approximations have been exploited in the inference world (before the popularization of the reparameterization trick)  for many years. Methods, where we replace a term in the joint distribution with a simpler function, are known in the variational inference world as local variational approximations, see, e.g. Murphy’s book (Machine Learning: A Probabilistic Perspective, 2012, Sec. 21.8) as a reference. The community has departed from such approaches as using the re-parameterization trick is unbiased, more general (e.g., not limited to Gaussian encoders) and allows for highly automated methods (no need to do derivations on a case-by-case basis).  Nevertheless, I encourage the authors to thoroughly explore the literature on variational inference with regards to these types of approximations. It may well be the case that, in the future, we revert back to these methods if they perform well in practice with modern architectures. For this, more comprehensive evaluations and comparisons are needed.